# The Serum/Glucocorticoid-Regulated Kinase 1 Is Targeted by miR-19a in CD4+ T Cells

**DOI:** 10.3390/cells12010133

**Published:** 2022-12-29

**Authors:** Julie Weidner, Carina Malmhäll, Vahid Arabkari, Aidan Barrett, Emma Boberg, Linda Ekerljung, Madeleine Rådinger

**Affiliations:** Krefting Research Centre, Institute of Medicine, Sahlgrenska Academy, University of Gothenburg, 40530 Gothenburg, Sweden

**Keywords:** microRNA, CD4+ cells, T cell polarization, asthma, gene expression

## Abstract

The polarization of CD4+ T cells into different T helper subsets is an important process in many diseases, including asthma. Part of the adaptive immune system, T cells are responsible for propagating signals to alert and prime the immune system. MicroRNAs (miRNAs) are small non-coding RNAs that act on numerous targets in the cell to regulate a variety of cellular processes, including roles in T cell polarization. In this study, we aimed to identify genes dysregulated in peripheral blood mononuclear cells from individuals with asthma. Moreover, we sought to examine miRNAs that may regulate the candidate genes and explore their functional relationship. Utilizing a focused gene array, we identified the serum/glucocorticoid-regulated kinase 1 (*SGK1*) gene to be upregulated in circulating peripheral blood mononuclear cells, which included T cells, from individuals with asthma. Several miRNAs were bioinformatically identified to target *SGK1*, but miR-19a was the only screened candidate that negatively correlated to *SGK1* expression. Further analysis of the miR-19a-*SGK1* relationship showed a negative correlation in CD4+ T cells in situ and direct binding in vitro during T cell activation. Moreover, we observed a negative correlation of miR-19a and *SGK1* during early type 2 polarization of CD4+ naïve human T cells. Thus, we suggest that miR-19a has a role in binding and regulating *SGK1* transcript levels during T cell development.

## 1. Introduction

Asthma is a chronic inflammatory disease of the airways that manifests in a heterogenous manner and includes multiple disease subgroups. The most common and well-studied type is allergic asthma, which is driven by a type 2 (T2) immune response characterized by secretion of cytokines, including interleukin-4, -5 and -13, and eosinophil recruitment [1]. This response relies on signals from multiple cell types, including T helper (TH) 2 cells, which instigate a type 2 signaling cascade. In patients with asthma, this aberrant T2 signaling is often treated using inhaled corticosteroids (ICS) to dampen the cellular immune response. The activation and polarization of TH cells have been extensively studied, and multiple TH cell types (such as TH1, TH2, TH17 and TH9) have been identified in human and murine models [2,3]. Each TH subtype activates its own unique signaling cascade, which can be used to characterize and identify various disease states. Once thought to terminally differentiate into a specific subclass of TH cell, recent studies have focused on the plasticity of TH cells and genetic, epigenetic and environmental factors that may influence TH cell activation and polarization.

Increasing evidence suggests that non-coding RNAs regulate numerous processes in the cell. MicroRNAs (miRNAs) are a set of non-coding RNAs that act as master regulators of post-transcriptional gene expression [4]. miRNAs generally bind to target sites within the 3′ UTR of mRNAs via an ~8 nucleotide seed sequence, which then leads to their downregulation either through translational repression or cleavage and decay. Furthermore, miRNAs can bind one or several different mRNA targets in a spatial and temporal manner, leading to a complex regulation that is currently not well understood. As miRNAs are generally extremely stable and ubiquitously expressed, their dysregulation has been shown in numerous studies to be involved in the pathogenesis of disease [2,4,5,6,7]. More recently, the involvement of miRNAs in inflammation, asthma and other airway diseases has been demonstrated [2,5,6,7]. Several studies have utilized miRNAs as potential circulating biomarkers to distinguish between diseases or disease subtypes [8,9,10], but mechanistic studies of miRNAs and the genes they target are lacking, especially in asthma.

Using a gene array, we found the serum/glucocorticoid-regulated kinase 1 (*SGK1*) gene to be upregulated in circulating peripheral blood mononuclear cells (PBMCs) from individuals with asthma. While Sgk1 has been implicated as an important factor in driving T cell polarity in murine models [11,12,13,14], very little is known about its role in humans and asthma [12,14]. Furthermore, recent studies have identified dysregulated miRNAs in T cells from individuals with asthma [15,16], leading us to question whether miRNAs may regulate *SGK1*. Thus, we aimed to determine if miRNAs regulated *SGK1* expression and if this regulation may play a role in T cell biology in humans. Herein, we describe the relationship between *SGK1* and miR-19a in human CD4+ T cells and suggest that miR-19a plays a role in binding and regulating *SGK1* transcript levels during T cell development.

## 2. Materials and Methods

### 2.1. Study Participants

Participants used in this study were recruited from the West Sweden Asthma Study cohort [17], as previously described [10]. Additional healthy individuals were recruited for the PrimeFlow™ RNA Assay and T cell polarization studies. Informed consent was obtained from all participants, and ethical approval was obtained from the Gothenburg County Regional Ethical Committee (no. 593-08 and 906-2016). Clinical characteristics of the participants are shown in Table 1.

### 2.2. Human PBMC Isolation and Treatment

Whole blood was collected in EDTA Vacutainer^®^ tubes (Becton Dickinson, Franklin Lakes, NJ, USA) for differential cell count and C-reactive protein measurements (Clinical chemistry, Sahlgrenska University Hospital, Gothenburg, Sweden), as well as PBMC isolation, as previously described [18]. Isolated cells were counted using a Sysmex XP-300 Automated Haematology Analyzer (Sysmex Europe, Norderstedt, Germany) and resuspended at a concentration of 2 × 10^6^ cells/mL. Approximately 2 million cells were seeded per well and grown in RPMI complete medium (RPMI 1640, 1% [*v/v*] L-Glutamine and 1% penicillin-streptomycin, all from Hyclone, Logan, UT, USA) and supplemented with 10% (*v/v*) heat-inactivated fetal bovine serum (Sigma-Aldrich, St. Louis, MO, USA) at 37 °C with 5% CO_2_. Cells were treated for 48 h with 0.2 µg/mL αCD3 (Purified NA/LE Mouse anti-Human CD3, clone UCHT1 (BD Pharmingen™, BD Biosciences, Franklin Lakes, NJ, USA)) and 0.33 µg/mL αCD28 (Purified NA/LE Mouse anti-Human CD28, clone CD28.2 (BD Pharmingen, BD Biosciences)), 2.5 µM dexamethasone (Merck, Branchburg, NJ, USA), either both treatments concurrently or medium only as control. Cells were collected and used in subsequent RNA or flow cytometry analyses.

### 2.3. Enrichment of CD4+ T Cells Post Treatment

PBMCs were harvested after 48 h of treatment, as described above, before being subjected to negative magnetic separation to enrich for CD4+ T cells according to the manufacturer’s instructions (MojoSort™ Human CD4 T cell Isolation Kit, Biolegend, San Diego, CA, USA). Sorted CD4+ T cells were collected in QIAzol^®^ lysis reagent (Qiagen Sciences, Germantown, MD, USA) and stored at −80 °C until further analysis.

### 2.4. Enrichment and Polarization of CD4+ Naïve T cells

PBMCs were isolated as described above, and CD4+ naïve (CD4+CD45RA+) T cells were isolated using the Magnisort™ Human CD4 Naïve T cell enrichment kit (Invitrogen by Thermo Fisher Scientific, Life Technologies Corp., Carlsbad, CA, USA) according to the manufacturer’s instructions. Isolated CD4+ naïve T cells were counted and resuspended at 2 × 10^6^ cells/mL in Complete T cell culture medium (RPMI complete with 1% MEM Non-essential amino acids (Sigma-Aldrich), 1% Sodium Pyruvate 100 mM (Gibco, Life Tech Limited, Paisley, UK), αCD3 and αCD28 each added at 1 µg/mL). Two million cells were plated per well and the appropriate 2× polarization medium was added to each well. The following culture media was prepared form the corresponding cells: TH0 was Complete T cell culture medium only, TH1 medium (2×) was αIL-4 (10 µg/mL; Ultra-LEAF™Purified anti-human IL-4 clone MP4-25D2, BioLegend^®^), rhIL-12 (20 ng/mL; Peprotech, UK) and αIL-10 (10 µg/mL; Ultra-LEAF™Purified anti-human IL-10 clone JES3-9D7, BioLegend^®^) in Complete T cell culture medium and TH2 medium (2×) was αIFNγ (10 µg/mL; Ultra-LEAF™Purified anti-human IFN-γ clone B27, BioLegend^®^); rhIL-4 (10 ng/mL; Peprotech), and αIL-10 (10 µg/mL) in Complete T cell culture medium. Cells were cultured at 37 °C with 5% CO_2_ for 5 days. On day 5, cells were allowed to rest using medium with rhIL-2 (50 U/mL; Peprotech) but without αCD3 and αCD28. On day 7, cells were restimulated with the initial medium until day 10–12. Cells were subsequently used for flow cytometry or stored in QIAzol^®^Lysis reagent until RNA isolation and analysis. For kinetic experiments, cells were collected on days 0, 3, 5, 7 and 10.

### 2.5. RNA Isolation

Cells were lysed in QIAzol^®^ Lysis Reagent and total RNA was isolated using miRNeasy Mini Kit (Qiagen) according to the manufacturer’s protocol, with the exception that Phasemaker™ tubes (Invitrogen by Thermo Fisher Scientific, Life Technologies Corp., Carlsbad, CA, USA) were utilized in the phase separation stage, as previously described [10]. RNA was measured using an Agilent RNA 6000 Nano kit on an Agilent 2100 Bioanalyzer running 2100 Expert Software (Agilent Technologies Inc.; Santa Clara, CA, USA) or a DeNovix Microvolume Spectrophotometer DS-11FX+ (DeNovix, Inc., Wilmington, DE, USA).

### 2.6. RT^2^ Profiler PCR Array

Total RNA (500 ng) was reverse transcribed to cDNA using the RT^2^ First Strand Kit (Qiagen) according to the manufacturer’s instructions. cDNA was mixed with RT^2^ SYBR^®^ green qPCR Mastermix (Qiagen) and nuclease-free water and pipetted into the RT^2^ Profiler Human Glucocorticoid Signaling array (Cat#PAHS-154Z, Qiagen) according to the manufacturer’s instruction and run in a CFX96 Touch Real-Time PCR Detection System (Bio-Rad Laboratories, Hercules, CA, USA). Data were analyzed via the GeneGlobe Analysis software available on the Qiagen homepage (www.qiagen.com, accessed on 7 October 2022).

### 2.7. mRNA and miRNA Expression Analysis Using RT-qPCR

Isolated total RNA from PBMCs was reverse transcribed to cDNA for subsequent mRNA analysis using the iScript™ cDNA Synthesis Kit (Bio-Rad) or for miRNA analysis using the miRCURY LNA RT Kit (Qiagen). Isolated total RNA from Jurkat pull-down experiments was reversed transcribed to cDNA using High-Capacity cDNA Reverse Transcription Kit (Applied Biosystems™, Thermo Fisher Scientific Baltics, Lithuania). All qPCR reactions were run on a CFX96 Touch Real-Time PCR Detection System (Bio-Rad) using SSO advanced SYBR Green (Bio-Rad) according to the manufacturer’s protocols. mRNA primers were purchased from Sigma-Aldrich, and miRNA expression was assessed using pre-designed miRCURY LNA miRNA PCR assays (Appendix A) and miRCURY LNA SYBR Green PCR kit (Qiagen) according to the manufacturer’s instructions.

### 2.8. PrimeFlow™ RNA Assay

The PrimeFlow™ RNA assay was performed as three separate experiments of 6 subjects in total (3 healthy and 3 asthmatic subjects) using a commercial assay kit (PrimeFlow™ RNA Assay, Invitrogen by Thermo Fisher Scientific, Affymetrix, Carlsbad, CA, USA). All buffers mentioned below were provided in the PrimeFlow™ RNA assay Kit. PBMCs were processed throughout the staining and hybridization procedure according to the manufacturer’s instructions (PrimeFlow™ RNA Assay with microRNA Pretreatment Protocol). In brief, PBMCs were incubated with 1 mg/mL of Human IgG (Sigma-Aldrich, St. Louis, MO, USA), followed by staining with viability dye (Live/Dead™ Fixable Aqua stain, Invitrogen, Life Technologies Corp., OR, USA) and surface antibody (αCD4). After the viability and surface staining, PBMCs were treated with PrimeFlow™ microRNA Pretreatment Buffer (Invitrogen), then fixed in Fixation buffer #1 before permeabilization with Permeabilization buffer with RNase inhibitor. Samples were then fixed in Fixation buffer #2, washed and kept overnight at 4 °C in wash buffer with RNase inhibitor. To detect cellular miRNA and mRNA, sequential hybridizations were performed in a dry incubator at 40 °C. Target probe sets were first hybridized, then signal amplification hybridization was performed using pre-amplifierDNA and amplifierDNA, followed by the corresponding fluorescent labeled probes (Type 1, Alexa Fluor 647; Type 4, Alexa Fluor 488). Probe sets used were the single pair target probe set for human miR-19a-3p (MIMAT0000073, VM1-10350-PF) and for human *SGK1* (VA4-3083904-PF). For each experiment, a control probe, human ribosomal protein L13A (NM_012423, VA4-13187-PF), was performed in one tube to verify the hybridization process. Following hybridization, amplification and labeling of the probes, intracellular staining was performed. Antibodies (αT-bet, αGATA-3, αRORγt and αFOXP3) diluted in permeabilization buffer were added and incubated for 45 min at 4 °C. Several steps of washing were performed with appropriate buffers throughout the staining and hybridization, following manufacturer protocols. Finally, cells were analyzed using a BD FACSVerse flow cytometer (BD Biosciences) running FACSuite software, collecting 0.5–1.6 × 10^6^ events/sample. Analysis of the data was performed using FlowJo Software (TreeStar, OR, USA). Only live, singlet CD4+ lymphocytes, considered CD4+ T cells, were analyzed. Gating of miR-19a, *SGK1* and surface/intracellular markers was determined using control samples by the fluorescence minus one (FMO) approach, i.e., controls containing all markers except the one of interest were used to set gates (Appendix A). Antibodies used are given in Appendix A. Median fluorescence index (MFI) values were used to determine expression levels of *SGK1* mRNA and miR-19a in the respective population (Appendix A).

### 2.9. Imaging of CD4+ T Cells Using PrimeFlow™ RNA Assay

As described above, PBMCs were harvested after 48 h of treatment, CD4+ T cells were isolated using the MojoSort™ Human CD4 T cell Isolation Kit and PrimeFlow RNA™ Assay was performed until the hybridization step. After hybridization, cells were washed several times, then spotted on poly-L-lysine coated slides, mounted with Prolong Diamond Antifade (Invitrogen) and stored at 4 °C in the dark until imaging. Imaging was performed on a Leica SP8 confocal microscope (Leica, Buffalo Grove, IL, USA). Mean intensity per cell (n > 55 per condition) was calculated using ImageJ (Bethesda, MD, USA).

### 2.10. RNA Pulldown

The protocol for RNA pulldown was adapted from Zhou et al. [19]. Biotinylated miRNA mimics were obtained for miR-19a (hsa-miR-19a-3p miRCURY LNA miRNA Mimic, MIMAT0000073: 5′UGUGCAAAUCUAUGCAAAACUGA, GeneGlobeID: YM00472044) and cel-miR-39-3p (MIMAT0000010: 5′UCACCGGGUGUAAAUCAGCUUG, GeneGlobeID: YM00479902, both from Qiagen), as a negative control. Jurkat cells (ACC 282 obtained from DSMZ, Braunschweig, Germany) were grown in RPMI complete medium at 37 °C and 5% CO_2_. Cells were treated (approximately 15 × 10^6^ cells per treatment), as previously described, for 48 h, then collected by brief centrifugation. Pelleted cells were lysed in 100 mM KCl, 5 mM MgCl_2_, 10mM HEPES pH 7.0, 0.5% NP-40/IPEGAL with 1 µL/100 µL rxn HALT protease and phosphatase inhibitor cocktail (Invitrogen) and RNaseOUT Recombinant Ribonuclease Inhibitor (40 U/µL; Invitrogen) on ice. Lysates were cleared by centrifugation and added to an equal amount of 2x TENT buffer (20 mM Tris pH 8, 2 mM EDTA, 500 mM NaCl, 1% Triton-X) with 200 µM biotinylated miRNA oligo and RNaseOUT, then mixed by gentle pipetting and incubated at room temperature for 45 min. Lysate reactions were added to 15 µL of magnetic Dynabeads (M-280 Steptavidin, Invitrogen) or Pierce Streptavidin Magnetic Beads (Pierce Biotechnology, Thermo Scientific, Rockford, IL, USA) pre-washed 2x with 0.1 M NaOH, 0.05M NaCl and 1x with 0.1 M NaCl and stored up to 2 days in 1x TENT buffer. Bead/lysate mixtures were then incubated for 45 min at room temperature with end over end rotation. Washing was performed twice using an equal amount of ice cold 1xPBS; the reactions were then placed on a magnet and flow through was discarded. Beads were resuspended in 0.1% SDS and incubated for 5 min at room temperature. Finally, an equal amount of 1xPBS was added, the bead solution was placed on the magnet and the entire supernatant was recovered. QIAzol^®^ was immediately added to the supernatant and stored at −80 °C until RNA isolation, as described, using an miRNeasy micro kit (Qiagen).

### 2.11. Statistics

Statistics are presented as box and whiskers plots where the box represents 25–75% and the whiskers 5–95% of the data, analyzed using a Mann–Whitney non-parametric test. Spearman correlation testing was used to determine correlations between miRNA and mRNA expression. PrimeFlow data were analyzed using a non-parametric statistical test (Friedman test ANOVA followed by Dunn’s multiple comparisons test). Statistical analyses were performed with GraphPad Prism 9 software (GraphPad Software Inc., San Diego, CA, USA). *p* < 0.05 was considered significant.

## 3. Results

### 3.1. The Glucocorticoid Signaling Pathway Is Altered in Asthma

Glucocorticoids, in the form of ICS, are among the most common treatments for asthma. We aimed to determine how the glucocorticoid signaling pathway may be altered in PBMCs from individuals with asthma compared to healthy individuals. We found that in freshly isolated PBMCs, roughly 30% of the genes examined were upregulated two-fold or greater in asthma as compared to healthy individuals (Figure 1A and Table 2). We then examined the connections between genes present in the array using high confidence STRING analysis [20]. When examining only genes with documented connections, a hub formed between the CAMP responsive element binding protein (CREB1), SGK1 and the glucocorticoid receptor (NR3C1) (Figure 1B). Recent studies have identified Sgk1 to be involved in controlling differentiation of TH or T regulatory (T reg) cell subsets [11,12,13,14]; therefore, we examined if this upregulation in *SGK1* could be validated. Indeed, upon performing qPCR analysis in additional subjects, we observed a significant increase in *SGK1* transcript levels between asthma and healthy individuals (Figure 1C).

### 3.2. miR-19a Is Predicted to Target SGK1

miRNAs are involved in numerous processes and act as master regulators in asthma pathogenesis [7]. To determine if miRNAs may be responsible for regulating the expression of *SGK1*, we searched for potential binding sites using miRNet, a comprehensive database that integrates findings from multiple databases [21]. We chose candidates to examine based on miRNet prediction and those previously reported to be associated with asthma or the glucocorticoid signaling pathway [10,16,22]. We found that miR-19a negatively correlated to *SGK1* in total treated PBMCs and displayed opposite expression patterns (Figure 2A,B). miR-19a was previously identified to be altered in airway cells in individuals with asthma [15,23,24]. When examining potential binding sites, we found that *SGK1* contains two conserved binding sites for miR-19a in the 3′UTR (Figure 2C). Moreover, when T cell transcription factors were examined in the same treated PBMCs, we found strong positive correlations to *GATA3* and *RORC* (Figure 2D). Together, these findings suggest that miR-19a may regulate *SGK1* in TH cells.

### 3.3. SGK1 and miR-19a Exhibit Opposing Expression Patterns in CD4+ T Cells

As both SGK1 and miR-19a have been reported to play roles in T cell development and activity, we wanted to determine if the relationship observed in whole PBMC culture was found in CD4+ T cells. PBMCs were treated with αCD3/αCD28 to stimulate T cells, dexamethasone or a combination of the two conditions. The expression of miR-19a, *SGK1* and T cell transcription factors was then examined using PrimeFlow technology [18]. *SGK1* expression was significantly increased in CD4+ T cells under αCD3/αCD28 stimulation (Figure 3B), whereas miR-19a expression was significantly decreased (Figure 3C). Similar opposing expression patterns were observed with dexamethasone and the combined treatment. A negative trend was observed between miR-19a and *SGK1* in CD4+ T cells (Figure 3D). We also observed increased expression of the transcription factors T-bet, GATA-3 and FOXP3 in CD4+ T cells under αCD3/αCD28 conditions (Appendix A). TH1, TH2 and Treg reflected what was seen in total CD4+ T cells, with *SGK1* intensity increasing and miR-19a decreasing under T cell activation conditions (Appendix A). RORγt was also examined, but very few cells were found to express this transcription factor (data not shown).

Furthermore, the expression of *SGK1* and miR-19a was visualized in CD4+ T cells that were sorted and stained using the PrimeFlow protocol. As in the PrimeFlow RNA assay, we observed increased expression of *SGK1* mRNA under αCD3/αCD28 stimulation and the combined conditions compared to Dex treatment (Figure 4 and Appendix A). As in flow cytometry, nearly every cell expressed miR-19a (red), but not every cell expressed *SGK1* (green) (Figure 3E and Appendix A).

### 3.4. miR-19a Binds SGK1

We now have evidence that miR-19a and *SGK1* are found in the same CD4+ T cells, that their expression tends to correlate with one another and that *SGK1* is predicted to contain two miR-19a binding sites in its 3′UTR. These findings motivated us to ask whether miR-19a could directly bind *SGK1*, thus downregulating its expression. Treated lysates from the CD4+ Jurkat cell line were incubated with biotinylated miR-19a and target binding was assessed (Figure 5A). Expression of the candidate target gene *SGK1* along with two genes that are not predicted targets of miR-19a, *GATA3* and *NR3C1*, were assessed. We found that miR-19a specifically bound *SGK1* (Figure 5B), but did not bind either of the off-target genes (data not shown).

### 3.5. SGK1 and miR-19a Levels Are Significantly Correlated in Early TH2 Cell Polarity

We have consistently observed that under conditions of T cell activation, *SGK1* expression is increased. This observation suggests that, as in mice, SGK1 may play a role in T cell polarization. To test this, we isolated CD4+ naïve T cells from whole blood and subjected them to TH1 or TH2 polarization conditions. After 10 days, cells polarized under TH1 polarization conditions expressed T-bet (Figure 6A and Appendix A), and TH2 polarized cells exhibited an increase in GATA-3 positive cells (Figure 6B and Appendix A), as expected. As the interaction between *SGK1* and miR-19a is likely quite transient, we performed a time course to examine expression kinetics at 0, 3, 5, 7 and 10 days (Figure 6C,D). Early in T cell polarization (<5 days), we observed that expression levels of miR-19a and *SGK1* appeared to negatively correlate. Despite having few subjects, we found increased miR-19a levels and decreased *SGK1* levels in both TH1 and TH2 cells on day 3 compared to day 0 (Figure 6E–G; Appendix A). Moreover, naïve CD4+ T cells treated with T2 polarizing conditions showed a significant negative correlation between miR-19a and *SGK1* after 3 days of polarization (Figure 6G; Appendix A). Thus, our data suggest that miR-19a targets *SGK1* during the early stages of T cell polarization, especially under T2 polarizing conditions.

## 4. Discussion

In recent years, miRNA regulation in airway and immune diseases has become a topic of increasing interest. Although the dysregulation of numerous miRNAs in circulation or locally in the lung have been identified [7], the mRNAs that they target and the effects of this targeting are still understudied. Therefore, we sought to determine genes dysregulated in the glucocorticoid signaling pathway in individuals with asthma compared to their healthy counterparts. Additionally, we wanted to identify and examine the relationship that miRNAs may have in regulating our candidate genes. Here, we observed that the gene *SGK1* was upregulated in individuals with asthma. Furthermore, we identified a miRNA previously reported to be associated with asthma pathogenesis, miR-19a, that was bioinformatically predicted to bind *SGK1*. We observed a negative relationship between *SGK1* and miR-19a in CD4+ T cells during conditions used to stimulate T cell activation in vitro, suggesting that miR-19a may be regulating *SGK1*. Furthermore, we found that there was direct binding of *SGK1* mRNA to miR-19a by pulldown assay. These data suggest that miR-19a interacts with *SGK1* in CD4+ T cells. Lastly, we observed that during polarization of naïve CD4+ T cells, there was a significant negative correlation between miR-19a and *SGK1* during early TH2 cell polarization. Collectively, our data suggest a direct regulatory relationship between *SGK1* and miR-19a in CD4+ T cells.

Murine studies have previously indicated a role for Sgk1 in T cell polarization, but to date, this role has not been examined in humans [11,12,13,14]. In a T cell-specific knock out of *Sgk1*, Heikamp and colleagues found that *Sgk1* was required for the proper polarization of TH2 cells. Furthermore, they found that T cell *Sgk1*^−/−^ mice did not respond to allergen challenge in a murine model of allergic asthma [12]. In line with these findings, we observed that *SGK1* was increased in circulating PBMCs, including T cells, from asthmatic individuals (Figure 1). Additionally, we found that in human subjects, *SGK1* levels indeed appeared to have a role in polarization of CD4+ naïve T cells to a TH2 cell phenotype during the early stages of T cell polarization. By focusing on the kinetics of the transcript, we observed that levels of *SGK1* fluctuated over the course of a 10-day polarization procedure, where the strongest correlation between *SGK1* and miR-19a was observed in early TH2 cell polarization (Figure 6E). Moreover, using our in vitro models of T cell activation and polarization, we were able to probe the kinetics of both the target transcript and regulator in CD4+ T cells (Figure 3A–C, Appendix A and Figure 6A–G). This insight into differential mRNA and miRNA expression will allow future studies to examine the points where changes in both the transcriptome and proteome may be optimal to study T cell polarization.

Most work on the role of SGK1 in T cell development has focused on its role in the TH17-Treg axis [11,13]. We also found that there was a strong positive correlation of the *SGK1* transcript to *RORC,* the gene encoding for the TH17 transcription factor RORγt (Figure 2D), in whole PBMC cultures. However, we were not able to further explore the TH17-SGK1 relationship due to small numbers of RORγt+ cells in our PrimeFlow assay and lack of optimal TH17 polarization conditions (data not shown). Most recently, Wu et al. identified Sgk1 as an enhancer of TH9 cell differentiation and examined its role in murine models of asthma [14]. Like previous studies, our findings suggest a role for SGK1 in asthma [12,14]. Though asthma has long been believed to be primarily T2-driven, multiple phenotypes have emerged in recent years, and many exhibit a non-T2 phenotype [2]. Our demonstration of the position of *SGK1* at the axis of T cell differentiation may indicate its role in the pathogenesis of multiple asthma phenotypes.

miR-19a belongs to the miR-17~92 cluster of miRNAs, and silencing the miR-17~92 cluster has been shown to impair TH2 cell responses [2,6,7]. Our data support this finding, suggesting a role for miR-19a during early T cell polarization. Furthermore, profiling of miRNA expression in human airway T cells showed that miR-19a is highly expressed in individuals with asthma [15]. We have also observed that miR-19a is expressed in circulating CD4+ T cells (Figure 3 and Figure 4). Interestingly, decreasing miR-19a activity by inhibitors reduced TH2 cytokine production by both human and mouse TH2 cells. miR-19a was further shown to promote TH2 cell activity through targeting *PTEN, SOCS1* and *A20*. Interestingly, mouse studies have demonstrated that the miR-17~92 cluster regulated inflammatory properties of innate lymphoid cell type 2 (ILC2) activity and revealed overlapping miRNA-regulated gene expression networks in ILC2s and TH2 cells [25].

In our study, we used a pre-determined panel of genes that focused on the glucocorticoid signaling pathway to identify differences between healthy and asthmatic individuals. While this method is relatively quick and provides easy-to-process data, it will not uncover all changes in either the pathway or the samples assayed. Similarly, we examined only a handful of miRNAs that were reported to play a role in asthma and/or glucocorticoid signaling [7,22]. In the future, it would be interesting to profile these subjects using a more in-depth screening technique such as single-cell or total RNA sequencing. In single-cell RNA sequencing, circulating naïve, helper and memory T cells could be directly isolated and examined on a cell-to-cell basis for *SGK1* expression in combination with T cell transcription factors. This type of screening could aide in our understanding of the gradients of T cell differentiation. Recent advances may soon allow for small RNA determination in single-cell RNA sequencing [26]. This development would provide the possibility to capture and analyze miRNA–RNA relationships on the single-cell level. Due to the complex regulatory nature of miRNAs, it is highly unlikely that miR-19a is the only miRNA that regulates SGK1. We observed that under different treatment conditions, miR-19a appears to bind differently to *SGK1* mRNA (Figure 5B). This could be the result of multiple binding sites being targeted in the *SGK1* 3′UTR or one of the binding sites being inaccessible during a particular treatment, potentially due to another RNA or protein binding to *SGK1*. Future studies are warranted to determine which 3′UTR binding site in *SGK1* is important in T cell activation/polarization.

Our findings suggest a role for SGK1 as a potential future target in asthma. As we observed that *SGK1* transcript levels were upregulated in individuals with asthma, this may alter the polarization of TH2 cells, leading to aberrant T2 signaling. Therefore, modulation of *SGK1* levels may be an alternative method to dampen the T2 immune response instead of the commonly used ICS treatment. However, a larger validation cohort would be required to more definitively determine if increased *SGK1* expression is specific to asthma or subgroupings of asthma. Moreover, the downstream impact of altering SGK1 levels in humans would have to be explored.

Although *SGK1* was found to be upregulated in individuals with asthma, *SGK1* and miR-19a in CD4+ T cells appear to have a common regulatory relationship, independent of the disease. Our initial screen was performed in total PBMCs; thus, an additional cell type(s) may at least partially contribute to the increased *SGK1* expression observed in cells from asthmatic individuals. Moreover, our treatment of PBMC cultures with αCD3/αCD28 may have simulated a strong T cell activation. This strong activation may have somewhat normalized how T cells react, making the differences between cells from healthy and asthmatic individuals negligible. Due to limitations in sample collection, we were not able to recruit additional asthmatic subjects to examine miR-19a and *SGK1* expression in polarized, CD4+ naïve T cells. However, it appears that the relationship of miR-19a and *SGK1* is present in individuals both with and without asthma, suggesting a general mechanism during T cell polarization. We recognize that the cohort for these supporting data is relatively small; however, we have consistent expression patterns of miR-19a and SGK1 in situ via PrimeFlow and in vitro during early polarization, strengthening the confidence in the data.

In conclusion, our data suggest that the regulation of *SGK1* via the binding of miR-19a occurs at early stages of T cell polarization. Thus, further studies examining *SGK1* and miR-19a are warranted to determine their precise role in T cell polarization and the development of asthma in humans.

## Figures and Tables

**Figure 1 cells-12-00133-f001:**
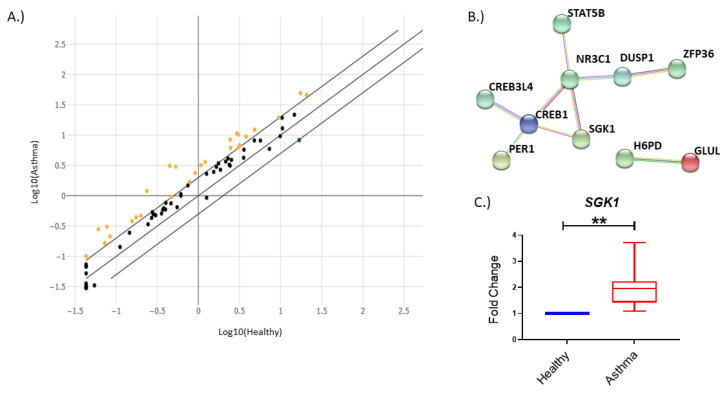
The glucocorticoid signaling pathway is altered in asthma. (**A**) Scatter plot showing distribution of up- and downregulated genes. (**B**) High-confidence (0.7) STRING network [20] of connected upregulated genes in individuals with asthma. (**C**) *SGK1* is significantly increased in PBMCs from individuals with asthma (n = 17). Shown as fold change from healthy individuals (n = 4), normalized to 1. ** = *p* < 0.01.

**Figure 2 cells-12-00133-f002:**
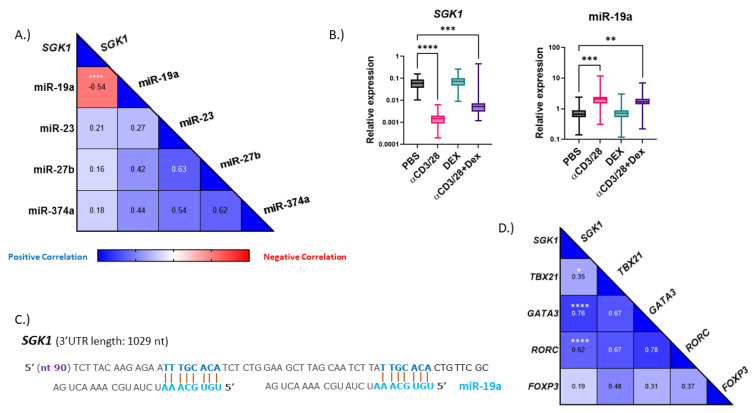
*SGK1* is predicted to be bound by miR-19a and correlates to T cell transcription factors. PBMCs were isolated from whole blood and treated for 48 h with αCD3αCD28 (αCD3/28), dexamethasone (DEX), αCD3αCD28 and dexamethasone (αCD3/28 + DEX) or medium alone (PBS), before RNA was isolated. (**A**) Bioinformatic analysis identified several miRNAs that may target *SGK1*; using pooled data from both healthy and asthmatic individuals, miR-19a was found to have a strong negative correlation. (**B**) Relative expression of *SGK1* and miR-19a from total PBMCs with the respective treatment. Relative expression shown as 2^−dCT^. n = 20/condition. (**C**) Predicted binding sites in the *SGK1* 3′UTR for miR-19a. Blue indicates the seed sequence. (**D**) Transcription factors *TBX21*, *GATA3*, *RORC* and *FOXP3* were examined in the same PBMCs to examine correlations to *SGK1*. * = *p* < 0.05; ** = *p* <0.01; *** = *p* < 0.001; **** = *p* < 0.0001.

**Figure 3 cells-12-00133-f003:**
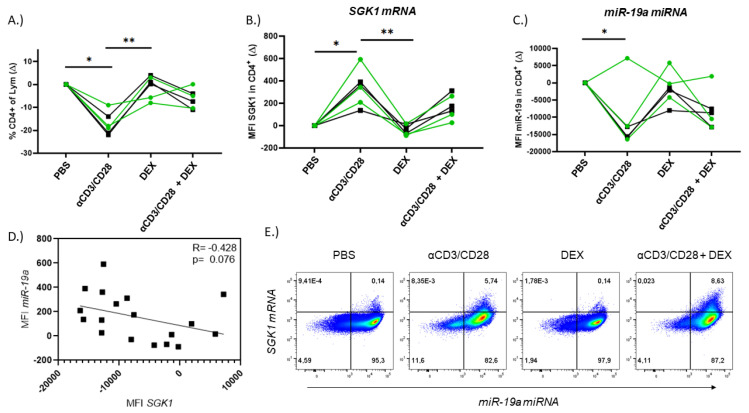
T cell activation causes opposing expression patterns of miR-19a and *SGK1* in CD4+ T cells. PBMCs were isolated from whole blood and treated for 48 h with αCD3αCD28 (αCD3/CD28), dexamethasone (DEX), αCD3αCD28 and dexamethasone (αCD3/CD28 + DEX) or medium alone (PBS) before performing PrimeFlow^®^ RNA Assay. (**A**) Percentage of lymphocytes that are CD4+. The intensity of *SGK1* mRNA (MFI) in CD4+ T cells (**B**). The intensity of miR-19a (MFI) in CD4+ cells (**C**). Correlation of MFI *SGK1* and MFI miR-19a in CD4+ T cells under treated conditions (**D**). Representative flow cytometry plot showing miR-19a and *SGK1* mRNA for each condition (**E**). A green line indicates an individual with asthma and black line indicates a healthy individual (n = 6). Expression is presented as a change from PBS control. MFI = median fluorescent intensity. Significance was determined by Friedman ANOVA. * = *p* < 0.05; ** = *p* < 0.01.

**Figure 4 cells-12-00133-f004:**
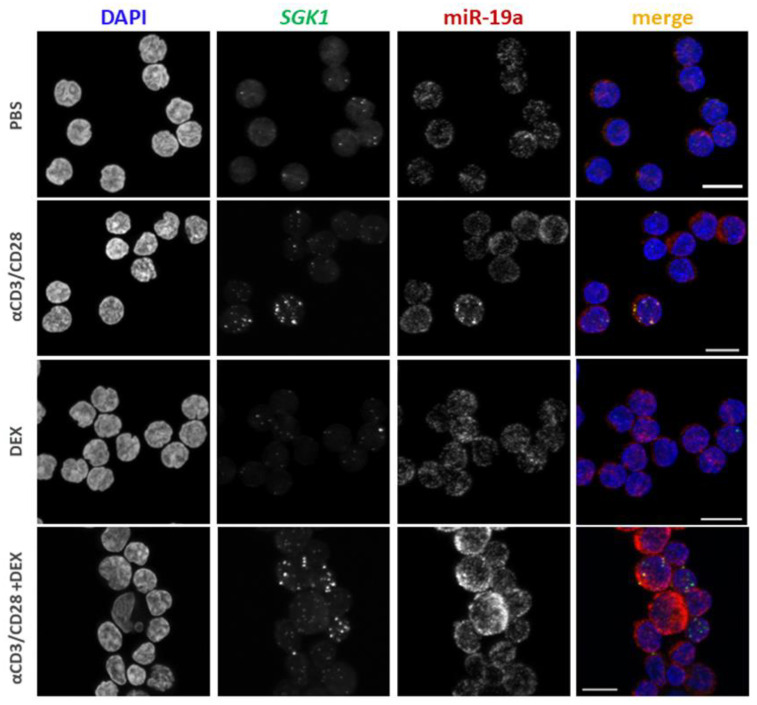
*SGK1* intensity is increased in αCD3αCD28 treated CD4+ T cells. Cells for imaging were hybridized with the PrimeFlow probes against *SGK1* mRNA and miR-19a miRNA and stained with DAPI to identify cell nuclei. Individual images for each probe and condition are shown: αCD3αCD28 (αCD3/CD28), dexamethasone (DEX), αCD3αCD28 and dexamethasone (αCD3/CD28+ DEX), medium alone (PBS vehicle). Representative images were selected for each condition. n = 3 individuals. Scale bar = 10 µM.

**Figure 5 cells-12-00133-f005:**
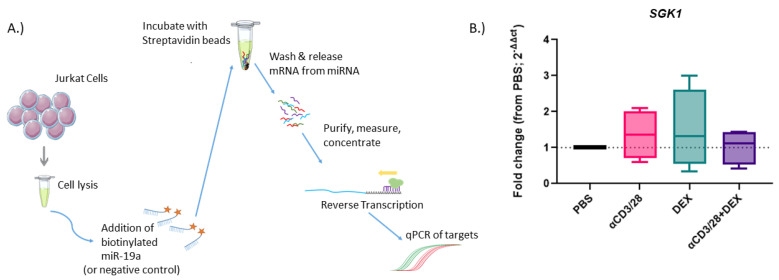
miR-19a directly binds *SGK1.* Biotinylated miR-19a was used to directly pull out *SGK1* from Jurkat cell lysates. (**A**) Graphical representation of the pulldown protocol. (**B**) *SGK1* mRNA binding was examined after pulldown by miR-19a. Results are shown as fold change from PBS (2^−ΔΔCT^); αCD3αCD28 (αCD3/28), dexamethasone (DEX), αCD3αCD28 and dexamethasone (αCD3/28 + DEX) or media alone (PBS vehicle).

**Figure 6 cells-12-00133-f006:**
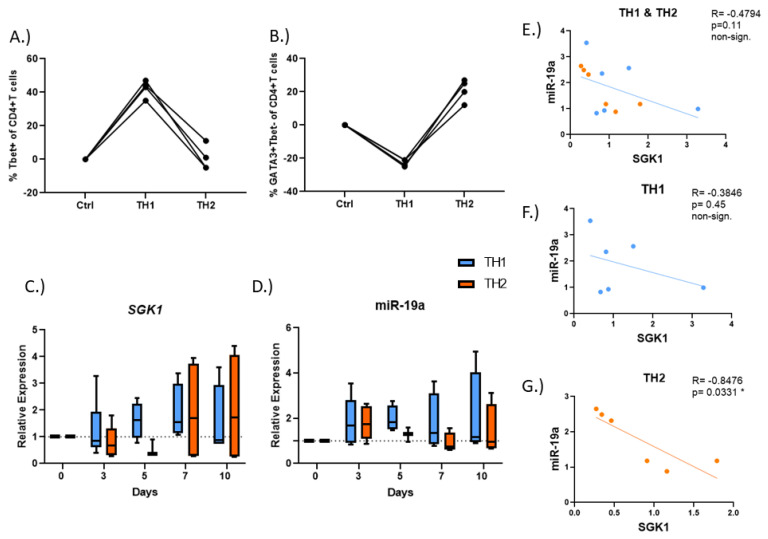
*SGK1* and miR-19a play a role in early TH2 cell polarization. CD4+ naïve T cells were isolated from whole blood and subjected to T cell polarizing conditions for up to 10 days. Cells stimulated under TH1 or TH2 conditions were examined after 10–12 days by flow cytometry, and T-bet (**A**) or GATA-3 (**B**) positive CD4+ T cells were measured. Expression of *SGK1* (**C**) and miR-19a (**D**) were examined on days 0, 3, 5, 7 and 10 while exposed to TH1 or TH2 polarizing conditions. Expression is normalized to *SGK1* and miR-19a expression in day 0 cells, and relative expression is shown as 2^−ddCT^. (**E**) Correlations of *SGK1* and miR-19a in TH1 and TH2 cells combined after 3 days. Correlations of *SGK1* and miR-19a in TH1 (**F**) and TH2 (**G**) cells. TH1 (blue)-T helper 1 cells; TH2 (orange)-T helper 2 cells. n = 3–6 individuals per timepoint. * = *p* < 0.05.

**Table 1 cells-12-00133-t001:** Demographic and clinical characteristics of study participants.

	Healthy	Asthma
Sex (M/F)	10/11	15/24
Age (years)	35 (29.5–48.5)	46 (35–55)
FEV1% p	101 (92–116) §	93 (84–105)
FEV1/FVC	0.83 (0.76–0.87) §	0.79 (0.74–0.81.5) #
Eosinophil count (10^9^ cells/L)	0.1 (0.095–0.2)	0.2 (0.1–0.4) **
Neutrophil count (10^9^ cells/L)	2.7 (2.35–3.4)	3.3 (2.8–3.5) *
Basophil count (10^9^ cells/L)	0 (0–0)	0 (0–0.1)
Monocyte count (10^9^ cells/L)	0.3 (0.3–0.4)	0.4 (0.3–0.5)
Lymphocyte count (10^9^ cells/L)	1.8 (1.55–2.15)	2.1 (1.8–2.3) *
Atopy (+/−)	0/11 §	18/21
ICS usage (+/−)	0/21	18/21

Data are shown as median with 25–75% in parentheses. FEV1 = forced expiratory volume in 1 s; %p = percent predicted; FVC = forced vital capacity; ICS = inhaled corticosteroid; § = 10 missing values. # = 2 missing values. Statistics are shown as significant from healthy. * *p* < 0.05; ** *p* < 0.01.

**Table 2 cells-12-00133-t002:** List of genes at least 2-fold up- or downregulated in asthma compared to healthy individuals, identified by RT^2^ profiler qPCR-based glucocorticoid signaling profiling array.

Symbol	Description	Fold Regulation (From Healthy Controls)
*ANXA4*	Annexin A4	2.98
*CEBPA*	CCAAT/enhancer binding protein (C/EBP), alpha	2.59
*CREB1*	CAMP responsive element binding protein 1	2.50
*CREB3L4*	CAMP responsive element binding protein 3-like 4	2.36
*DDIT4*	DNA-damage-inducible transcript 4	2.54
*DUSP1*	Dual specificity phosphatase 1	2.24
*GDPD1*	Glycerophosphodiester phosphodiesterase domain containing 1	2.53
*GLUL*	Glutamate-ammonia ligase	2.06
*H6PD*	Hexose-6-phosphate dehydrogenase (glucose 1-dehydrogenase)	2.02
*HNRNPLL*	Heterogeneous nuclear ribonucleoprotein L-like	5.71
*MT1E*	Metallothionein 1E	2.34
*NR3C1*	Nuclear receptor subfamily 3, group C, member 1 (glucocorticoid receptor)	2.04
*PDCD7*	Programmed cell death 7	2.17
*PER1*	Period homolog 1 (Drosophila)	4.02
*PLEKHF1*	Pleckstrin homology domain containing, family F (with FYVE domain) member 1	2.44
*RASA3*	RAS p21 protein activator 3	2.13
*RGS2*	Regulator of G-protein signaling 2, 24kDa	2.82
*SESN1*	Sestrin 1	6.99
*SGK1*	Serum/glucocorticoid-regulated kinase 1	3.66
*SLC19A2*	Solute carrier family 19 (thiamine transporter), member 2	5.08
*SNTA1*	Syntrophin, alpha 1 (dystrophin-associated protein A1, 59 kDa, acidic component)	2.27
*SPHK1*	Sphingosine kinase 1	4.61
*STAT5B*	Signal transducer and activator of transcription 5B	2.55
*TBL1XR1*	Transducin (beta)-like 1 X-linked receptor 1	3.47
*TNFAIP3*	Tumor necrosis factor, alpha-induced protein 3	3.35
*TSC22D3*	TSC22 domain family, member 3	−2.03
*ZFP36*	Zinc finger protein 36, C3H type, homolog (mouse)	2.49
*ZNF281*	Zinc finger protein 281	2.97

Red indicates upregulation and green indicates downregulation.

## Data Availability

Not applicable.

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
