# Peer review of "The Serum/Glucocorticoid-Regulated Kinase 1 Is Targeted by miR-19a in CD4+ T Cells"

_cells, 2022, doi:10.3390/cells12010133_

Round 1

Reviewer 1 Report

Weidner et al present a manuscript showing an upregulation of Serum/Glucocorticoid regulated Kinase 1 (SGK1) in human T cells from Asthma patients. Furthermore, the MicroRNA (miR-19a), which was previously reported to be associated with asthma pathogenesis, to negatively correlate with the SGK1 expression. In this line, Weidner et al show in situ and in vitro that miR-19a directly binds to the mRNA of SGK1. Interestingly, during early Type 2 polarization in human T cells miR-19a is expression in significantly decreased while SGK1 is significantly increased.

All in all, the data presented suggests a direct regulatory relationship between SGK1 and miR-19a which occurs at already during early stages TH2 cell development.

The manuscript is very well written and structured.

There are two major points though:

Figure 5B: This elegant experiment shows nicely the binding of miR-19a to SGK1 in Jurkat cells. However, there is just one data point depicted for each condition, does this mean N=1? If yes, at least N=3 experiments have to be done. In addition, also the statistical analysis should be done afterwards.

Figure 4: The representative cells are convincing, however, a statistical analysis of the correlation of all acquired pictures has to be performed. A further, minor point is that the scale of the scale bar is missing in the figure legend. Please add this.

Minor points:

Harmonize the Figures were possible, for example use the same color-code in all Figures throughout the manuscript (e.g. black  = healthy and green = asthma patients; TH1 = light blue and TH2 = orange).

Furthermore, use similar graphs for your data were possible, sometimes it is Box-Whisker, sometimes Dot-Plots and these are sometimes connected and sometimes not.

P5 Line 204 -210 Methods “Imaging”

To be more correct please change the section name to “Fluorescence in situ hybridization”

Furthermore, the methods description is very short, please add for example the sequence of the probes used.

Figure 3: I would suggest to simplify the figure and just show the data you are actually referring to in the text. I would just show A.), E.) and I.) of the left panel and M.) and N.) and the rest could be included in a supplementary Figure.

Figure 6 Please add the R value to the correlations

Author Response

Reviewer 1:

Weidner et al present a manuscript showing an upregulation of Serum/Glucocorticoid regulated Kinase 1 (SGK1) in human T cells from Asthma patients. Furthermore, the MicroRNA (miR-19a), which was previously reported to be associated with asthma pathogenesis, to negatively correlate with the SGK1 expression. In this line, Weidner et al show in situ and in vitro that miR-19a directly binds to the mRNA of SGK1. Interestingly, during early Type 2 polarization in human T cells miR-19a is expression in significantly decreased while SGK1 is significantly increased.

All in all, the data presented suggests a direct regulatory relationship between SGK1 and miR-19a which occurs at already during early stages TH2 cell development.

The manuscript is very well written and structured.

We thank the reviewer for their feedback on the writing and structure of our manuscript.

There are two major points though:

Figure 5B: This elegant experiment shows nicely the binding of miR-19a to SGK1 in Jurkat cells. However, there is just one data point depicted for each condition, does this mean N=1? If yes, at least N=3 experiments have to be done. In addition, also the statistical analysis should be done afterwards.

Thank you for the comment. We have now repeated the miR-19a-SGK1 binding experiment (n=4) in Jurkat cells (Fig 5B) as performed previously. There was specific binding of SGK1 to miR-19a as neither off target gene was above the detection limit by qPCR. There is a clear trend towards the increased binding of SGK1 to miR-19a under both αCD3/CD28 and Dex conditions as seen previously, however the increased binding under these conditions were not significant from PBS. 

Figure 4: The representative cells are convincing, however, a statistical analysis of the correlation of all acquired pictures has to be performed. A further, minor point is that the scale of the scale bar is missing in the figure legend. Please add this. 

We thank the reviewer for pointing out this aspect of our imaging. The main point of the imaging was to examine the cells which were the main subject of our PrimeFlow analysis (CD4+ T cells) and determine if we could use the same probe sets for visualizing the mRNA and miRNA target.  However, we realize we did make a statement regarding intensity. We have now quantified the mean intensity of at least 50 cells per condition (n=3 individuals) and these graphs are now located in Fig S3.

Thank you for noting the missing size of the scale bar (10 µM). This has now been added to the legend of Figure 4 (Line 349). 

Minor points:

Harmonize the Figures were possible, for example use the same color-code in all Figures throughout the manuscript (e.g. black  = healthy and green = asthma patients; TH1 = light blue and TH2 = orange). 

Furthermore, use similar graphs for your data were possible, sometimes it is Box-Whisker, sometimes Dot-Plots and these are sometimes connected and sometimes not.

We agree that the different graph styles was a bit confusing when looking through the paper.  We have attempted to harmonized this aspect by having the majority of graphs as box-whisker graphs and tried to maintain similar color schemes.  However, we have chosen to keep the PrimeFlow experiment (Fig 3 & FigS2) as line graphs to represent the differences that individual donors can have in this particular methodology. And to indicate that there were, overall, no major differences between healthy individuals and individuals with asthma.

P5 Line 204 -210 Methods “Imaging”

To be more correct please change the section name to “Fluorescence in situ hybridization”

Furthermore, the methods description is very short, please add for example the sequence of the probes used.

Thank you for the comment.  The imaging is performed in exact the same way as the PrimeFlow staining itself with all the same washes and probes (Lines 171-192). The difference being that we pre-sorted CD4+ cells using the MACs CD4+ sorting system, as described on line 114-119, as we wanted to enrich the population for imaging since we were not able to have a staining panel as complex as for the flow cytometer.

Due to the protocols for the PrimeFlow and imaging cells being the same, we felt it was not necessary to re-write them in the Imaging section. We having however updated the title of the section to “Imaging of CD4+ T cells using PrimeFlowTM RNA assay” (Line 207) and have tried to be more clear regarding the steps taken prior to imaging (Lines 208-214).

“As described above, PBMCs were harvested after 48 hrs of treatment, CD4+ T cells were isolated using the MojoSort™ Human CD4 T cell Isolation Kit and PrimeFlow RNA™ Assay was performed until the hybridization step. After hybridization, cells were washed several times, then spotted on poly-L-lysine coated slides, mounted with Prolong Diamond Antifade (Invitrogen) and stored at 4°C in the dark until imaging. Imaging was performed on a Leica SP8 confocal microscope (Leica, IL, USA). Mean intensity per cell (n>55 per condition) was calculated using ImageJ (MD, USA).”

Figure 3: I would suggest to simplify the figure and just show the data you are actually referring to in the text. I would just show A.), E.) and I.) of the left panel and M.) and N.) and the rest could be included in a supplementary Figure. 

We thank the reviewer for the suggestion. We agree that it makes sense to focus on the data that is most discussed. Thus, we have simplified Fig 3 to include the suggested data (relabeled A-E) and moved the specific T-cell transcription factor PrimeFlow data to Figure S2.

Figure 6 Please add the R value to the correlations 

Thank you for the suggestion. The R values have been added to the Fig 6 as requested.

Reviewer 2 Report

* The authors made reasonable efforts to reveal the targeting of SGK1 by miR-19a. However, the importance of this study is not evident. What are the future clinical studies based on your detection? 

* The language of your manuscript is not easy to understand so the style of your manuscript needs extensive modification. For example, your abstract contains many sentences belonging to the introduction and the work's aim is near your abstract's end. Only two background sentences in the abstract and the main abstract body belong to the main used methods and results.

* Please, mention all catalog numbers for all used devices and kits.

Author Response

Reviewer 2:

* The authors made reasonable efforts to reveal the targeting of SGK1 by miR-19a. However, the importance of this study is not evident. What are the future clinical studies based on your detection? 

Thank you for the comment on our manuscript. We have added a few sentences on the clinical impact and potential future studies that may be possible with the data we have provided (Line 476-482):

“Our findings suggest a role for SGK1 as a potential future target in asthma. As we observed that SGK1 transcript levels were upregulated in individuals with asthma, this may alter the polarization of TH2 cells leading to aberrant T2 signaling. Therefore, modulation of SGK1 levels may be an alternative method to dampen the T2 immune response instead of the commonly used ICS treatment. However, a larger validation cohort would be required to more definitively determine if increased SGK1 expression is specific to asthma or subgroupings of asthma. Moreover, the downstream impact of altering SGK1 levels in humans would have to be explored.”

* The language of your manuscript is not easy to understand so the style of your manuscript needs extensive modification. For example, your abstract contains many sentences belonging to the introduction and the work's aim is near your abstract's end. Only two background sentences in the abstract and the main abstract body belong to the main used methods and results. 

We apologize if the reviewer found the language and structure of our manuscript difficult to understand. We have sent the manuscript for language correction and editing and hope that it is now suitable for publication.

* Please, mention all catalog numbers for all used devices and kits. 

We appreciate the reviewer asking for catalog numbers on devices and kits, however, we feel that this knowledge is not appropriate to add as catalog numbers for laboratory items and kits vary by region.  We have, however, made sure that all items used in this manuscript have a supplier listed, as well as more detailed information for all probes, flow cytometry antibodies and qPCR primers used.

Round 2

Reviewer 1 Report

Most of the raised points were addressed or discussed after the revision.